# Violence Against Women on Social Networks: A Descriptive Analysis

**DOI:** 10.3390/healthcare13202574

**Published:** 2025-10-14

**Authors:** Pedro José López-Barranco, Samara López-Yepes, María Belén Conesa-Ferrer, Pedro Simón Cayuela-Fuentes, María del Mar Beladiez-Pérez, Ismael Jiménez-Ruiz

**Affiliations:** 1Faculty of Nursing, University of Murcia, 30120 Murcia, Spain; 2Murcian Institute of Biosanitary Research (IMIB), 30120 Murcia, Spain; 3Independent Researcher, 30100 Murcia, Spain

**Keywords:** social networks, violence against women, online violence, sexual violence

## Abstract

**Objectives**: This study aimed to identify the prevalence of gender-based violence experienced through social networks among adult women in Spain. Specific objectives included describing in-person sexual violence within the context of GBV and analyzing the relationship between GBV experienced on social networks and in-person sexual violence. **Methods**: This observational, cross-sectional, and correlational study surveyed 1177 adult women aged 18–59 years. Data were collected through validated instruments, including the Cyber Dating Abuse Questionnaire, Online Sexual Victimization Scale, and Dating Violence Questionnaire. Statistical analyses, including the Mann–Whitney U test, Kruskal–Wallis test, and Spearman’s Rho, were used to examine violence as a function of sociodemographic variables, social network usage, and pornography consumption. **Results**: Of participants, 68.2% reported experiencing GBV on social networks, 62.7% reported online sexual violence, and 66.0% reported in-person sexual violence. Gender-based violence was significantly correlated with online sexual violence (r = 0.390, *p* < 0.001) and in-person sexual violence (r = 0.463, *p* < 0.001). Women from lower socioeconomic backgrounds reported higher victimization rates for all forms of violence analyzed (*p* < 0.05). Increased daily social network usage and pornography consumption were associated with higher victimization rates (*p* < 0.05). **Conclusions**: Gender-based violence on social networks is pervasive among adult women in Spain and is closely linked to in-person sexual violence. Socioeconomic factors, time spent on social networks, and pornography consumption were key predictors of victimization. These findings highlight the need for targeted interventions addressing online violence to mitigate gender-based violence in broader contexts.

## 1. Introduction

Article 3 of the 2011 Istanbul Convention defines violence specifically against women as “a violation of human rights and a form of discrimination against women and shall mean all acts of gender-based violence that result in, or are likely to result in, physical, sexual, psychological or economic harm or suffering to women, including threats.” The same article makes specific reference to gender-based violence (GBV), which is understood as any “violence that is directed against a woman because she is a woman or that affects women disproportionately” [1]. The Spanish Government has defined GBV in a similar way to the Istanbul Convention, identifying this form of violence as “the violence exercised against women by their present or former spouses or by men with whom they maintain or have maintained analogous affective relations, with or without cohabitation, as an expression of discrimination, the situation of inequality and the power relations prevailing between the sexes” [2].

GBV is a global phenomenon affecting all societies, with victimization rates reaching truly disturbing levels. The World Health Organization (WHO) estimates that one in three women worldwide has experienced some form of physical and/or sexual violence at least once in her lifetime [3]. The 2022 European Survey on Gender-Based Violence (EU-GBV) found that 28.7% of all Spanish women—over four million—had experienced some form of GBV [4]. Among these forms of violence against women, psychological violence is the most frequently reported, experienced by 27.8% of all Spanish women. This is followed by physical violence, experienced by 12.7% [4]. Finally, the data on sexual violence show that 6.7% of the total population of Spanish women have been victims of this type of violence [4]. Intimate partner violence suffered through social media and the internet has emerged as a new form of violence with its own characteristics that are interesting to address [5], this new form of violence transcends distance and time, and can be perpetrated and suffered from anywhere in the world and for a longer period of time [6].Gender-based violence in the digital space is primarily experienced by women, with studies on young and adolescent populations revealing victimization rates as high as 21.3% [7]. One study in the Spanish adult population between the ages of 18 and 60 found that 41.6% of women in the sample surveyed had experienced gender-based violence in the digital environment [8].

All forms of gender-based violence have an impact on women’s health, with femicide being the most serious. From 1 January 2003 to June 2024, 1258 femicides were recorded in Spain as a result of gender-based violence [9]. In addition to the number of victims and fatalities, this phenomenon also has an impact on women’s health. Gender-based violence affects all areas of a person’s life: physical, psychological, sexual, reproductive, and social [10]. These consequences are also interrelated [10]. Similarly, it is important to note that although the effects of psychological violence are less visible, it is a major risk factor for later exposure to more severe forms of violence, such as extreme physical violence, sexual violence, or even femicide [11,12].

The forms of GBV and the channels through which they are exercised and suffered have been updated. With the rise of information and communication technologies, social networks have emerged as a new arena for perpetuating different forms of GBV, thereby broadening its scope. The risk is not limited to in-person exposure, but the emergence of new forms of interaction and communication, such as social networking sites, has changed the landscape in which this problem occurs [13,14,15].

Likewise, recent research has highlighted the need to consider pornography consumption as a potential factor associated with gender-based violence. Frequent exposure to pornographic content—particularly violent or degrading material—has been linked to sexist attitudes, the acceptance of rape myths, and a higher likelihood of perpetrating sexual or intimate partner violence [16,17]. These findings reinforce the importance of including this variable in the analysis of factors contributing to the perpetuation of violence against women.

In light of the issues described and their analytical significance, this study aims to identify the gender-based violence experienced via social networks by adult women in Spain. Specifically, it seeks to describe the in-person sexual violence occurring within the context of gender-based violence and analyze the relationship between such violence and the experiences of abuse facilitated through social networks.

## 2. Materials and Methods

### 2.1. Design and Procedure

This is an observational, descriptive, cross-sectional, and correlational study.

### 2.2. Participants and Sample Size

Study participants were adult and young adult women aged 18 to 59 years living in Spain. Age stratification was done in two groups, young adult women aged 18 to 24 years and adult women aged 25 to 59 years [18]. The sample was selected by non-probability and purposive sampling. Data collection was conducted online. Given the sensitive nature of the scale used, the electronic format was considered the most appropriate option to ensure the anonymity of responses. An attractive and user-friendly questionnaire was designed using the digital platform “Encuestas UM.” This format enabled distribution through a message sent to various university institutions and associations specialized in the subject. The message included a link to the questionnaire, along with information about the study and the informed consent form

Participants were required to meet a number of criteria: they must have been in an intimate or dating relationship, be of legal age, and agree to participate in the research on an anonymous basis. As the main objective of the study is to address gender-based violence experienced through social networks, only participants with a heterosexual orientation were selected. This decision stems from the definition of gender-based violence described by the Spanish government in 2004, which states that this form of violence is suffered only by women.

Prior to completing the survey, participants were informed of the purpose of the study and their right to withdraw from the questionnaire at any time during the completion process. Participants were assured of complete anonymity throughout the study.

### 2.3. Ethical Considerations

All participants participated with informed consent in writing, participants gave written consent prior to the start of the questionnaire, data were collected anonymously, participants were of legal age, free and autonomous in their responses, and the research was approved by the corresponding assigned committee.

The study adhered to the national and international guidelines of the Declaration of Helsinki, the European Data Protection Guidelines and the Law 3/2018 on Data Protection and Digital Rights [19,20]. The research study was approved by the Ethics Committee of the University of Murcia with registration number ACTA5/2024/IEC. All participants in the study were of legal age and gave their consent before completing the questionnaire. Data collection was carried out from 1 March 2024 to 3 June 2024.

### 2.4. Study Variables Analyzed

Participants were asked to provide information about their autonomous community, educational background, socioeconomic status, sexual orientation, year of birth, the social network they used most, the amount of time they spent on that social network each day, and whether and how often they used online pornography. Participants were asked about online violence experienced during their dating relationships, as well as sexual violence experienced both online and in person.

### 2.5. Instruments Used

Violence experienced online: Violence experienced through social networks in the context of intimate relationships was analyzed using the Cyber Dating Abuse Questionnaire (CDAQ) [21]. The CDAQ is capable of assessing technology-related violence experienced in the context of an intimate or dating relationship. It consists of 20 items, each of which is dual in nature and can measure both victimization and perpetration. Both victimization and perpetration of this form of violence are assessed on two dimensions: Direct Aggression and Control. The Direct Aggression dimension refers to behaviors experienced by the victim, such as humiliation, threats, and identity theft, that were intended to cause harm (e.g., My partner or ex-partner has threatened to physically harm me through new technologies). The Control dimension concerns the controlling behaviors experienced in relationships (e.g., My partner or ex-partner has kept track of my social media status updates). The CDAQ uses a six-point Likert-type response scale: 1 (this has never happened in our relationship), 2 (not in the last year, but it has happened in the past), 3 (rarely: it has happened one or two times), 4 (sometimes: it has happened between three and 10 times), 5 (frequently: it has happened between 11 and 20 times), and 6 (usually: it has happened more than 20 times). Given the research question and objectives, the items corresponding to the forms of violence experienced were used. The reliability indices of the questionnaire for the study sample were as follows: CDAQ overall; α = 0.942, Aggression-Direct; α = 0.855; and Aggression-Control; α = 0.946.

Sexual violence experienced online: Sexual violence experienced through social networks was assessed using the Online Sexual Victimization Scale (OSV) [22]. The OSV is made up of 10 items (e.g., Someone has threatened or coerced you to share erotic or sexual information about yourself). Responses are given on a five-point Likert-type scale: 0 (never), 1 (one or two times), 2 (three or four times), 3 (five or six times), and 4 (seven or more times). The reliability index for the questionnaire in the study sample was as follows: OSV; α = 0.881

In-person sexual violence experienced: To identify in-person sexual violence, we used the Dating Violence Questionnaire for Victimization and Perpetration (DVQ-VP) [23]. The DVQ-VP is capable of measuring different forms of violence experienced and perpetrated in relationships. For this study we used the dimension corresponding to the sexual violence experienced in relationships (e.g., Insists on touching you in ways that you do not like and do not want). The response options are based on a three-point Likert-type frequency scale: 0 (never), 1 (sometimes), and 2 (often). The reliability index for the questionnaire in the study sample was as follows: DVQ-V; α = 0.859.

### 2.6. Data Analysis

Analyses were performed using SPSS (Statistical Package for the Social Sciences) v 28.0 software [24].

Cronbach’s alpha (α) was determined for each of the instruments used. The Kolmogorov–Smirnov (K-S) statistical test was applied to analyze the distribution of the sample. Absolute frequencies (n), relative frequencies (%), means (M), medians (Me), and standard deviations (SD) were used for the descriptive analyses.

The study variables were not normally distributed in the sample according to the Kolmogorov–Smirnov test (*p* < 0.001). Due to the sample distribution, non-parametric tests were used in the subsequent analyses. The Mann–Whitney U non-parametric test was used to examine the forms of violence experienced as quantitative variables in relation to the dichotomous qualitative variables of age group and pornography consumption.

The Kruskal–Wallis non-parametric test was used to analyze the violence experienced as a quantitative variable, as a function of the following qualitative variables with more than two groups: socioeconomic status and daily social network usage frequency.

Spearman’s Bilateral Correlation Index (Spearman’s Rho) was applied to examine whether there was any correlation between the different forms of violence analyzed.

The results were considered statistically significant with a confidence level of 95% (*p* < 0.05).

## 3. Results

### 3.1. Sociodemographic Characteristics

A total of N = 1707 women initially responded to the questionnaire. After selecting women who were of legal age, adult or young adult, had given informed consent to participate in the study, were in or had been in an intimate or dating relationship, and whose sexual identity was heterosexual, the final number of women included in the analysis was n = 1177. The mean age of the participants was M = 33.2 years; SD = 9.8. The sociodemographic characteristics can be found in Table 1.

The study participants were drawn from various regions across the country, ensuring representation of diverse geographic areas. Additionally, they encompassed different age groups, corresponding to the target demographic of the research, which included both young adults and adults. In terms of social network usage, 7.5% of participants used social networks less than one hour per day, 60.9% used them between one and three hours per day, 26.0% used them between three and five hours per day, and 5.6% used them more than five hours per day. The most commonly used social network among the participants was WhatsApp, followed by Instagram, Facebook, Tiktok, and lastly Telegram.

With regard to pornography, 19.6% of the participants stated that they consumed pornography. Of those who reported consuming pornography, 26.2% did so once or several times a year, 52.6% once or several times a month, 17.7% once or several times a week, and 3.4% consumed it daily. Table 1 shows the frequencies of social network usage and pornography consumption among the participants.

### 3.2. Violence Experienced on Social Networks in Intimate Relationships

Of all the participants surveyed, 68.2% reported having experienced some form of violence in the context of an intimate or dating relationship through social networks. The Direct Aggression subscales revealed that 42.2% had experienced some form of direct aggression from their partner through social networks and 65.0% had experienced violence in the form of controlling behavior from their partner through social networks. A total of 62.7% of participants reported having been subjected to online sexual violence by their partners. Some 66.0% of the participants stated that they had been victims of in-person sexual violence at the hands of their partners. Table 2 shows the relative frequencies of the different forms of violence experienced.

### 3.3. Violence Experienced as a Function of Age Group

Among the different forms of violence experienced as a function of age group, significant differences were found only for in-person sexual violence, with adult women reporting higher levels of victimization compared to young adult women (U (95% CI) = 112,929.5; *p* < 0.05). Table 3 shows the data for the different forms of violence experienced, categorized by age group.

### 3.4. Violence Experienced as a Function of Socioeconomic Status

Differences were found for direct online violence, with women from lower socioeconomic backgrounds experiencing higher levels of victimization for this form of violence compared to women from middle and higher socioeconomic backgrounds (Kruskal–Wallis (95% CI) = 7.5; *p* = 0.024). Similar results were found for online sexual violence (Kruskal–Wallis (95% CI) = 11.1; *p* < 0.005) and in-person sexual violence (Kruskal–Wallis (95% CI) = 13.7; *p* < 0.001). Participants from lower socioeconomic backgrounds reported higher levels of victimization for these forms of violence compared to participants from other socioeconomic groups (see Table 4).

### 3.5. Violence Experienced as a Function of Time Spent Using Social Networks

Significant differences were found in the scores for violence experienced in the different groups categorized by the amount of time spent using social networks on a daily basis.

In terms of overall online violence, the group of participants who used social networks for more than five hours per day had higher levels of victimization for this form of violence (Kruskal–Wallis (95% CI) = 10.04; *p* = 0.018). On the subscale measuring the control dimension of online violence, the group using social networks for more than five hours per day scored higher on victimization (Kruskal–Wallis (95% CI) = 19.93; *p* = 0.000). Online sexual violence was experienced to a greater extent by the group using the social networks for between three and five hours per day (Kruskal–Wallis (95% CI) = 13.36; *p* = 0.001). In-person sexual violence was more prevalent among participants who used social networks for more than five hours per day (Kruskal–Wallis (95% CI) = 23.95; *p* = 0.000). Table 5 shows the findings for violence experienced as a function of the number of hours per day spent using social networks.

### 3.6. Violence Experienced as a Function of Pornography Consumption

Participants who reported using pornography demonstrated higher levels of victimization for overall online violence (U (95% CI) = 99,649.5; *p* < 0.05), direct online violence (U (95% CI) = 99,048.5; *p* < 0.05), and online sexual violence (U (95% CI) = 93,514.5; *p* < 0.05). Table 6 shows the results of violence experienced as a function of pornography consumption.

### 3.7. Correlation Analysis Between the Different Forms of Violence Experienced

There are correlations between the three main forms of violence analyzed. Overall violence experienced through social networks in relationships was significantly correlated with both online sexual violence (Spearman’s Rho = 0.390; *p* < 0.001) and in-person sexual violence (Spearman’s Rho = 0.463; *p* < 0.001). The strongest relationship found between the different variables was the correlation between online and in-person sexual violence (Spearman’s Rho = 0.518; *p* < 0.001). Table 7 shows the relationships between the different forms of violence analyzed.

## 4. Discussion

The study identified the levels of GBV experienced through social networks in a sample of adult women in Spain, as well as the levels of online and in-person sexual violence experienced by these women. The relationship between different forms of online and in-person sexual violence experienced through social networks was also explored.

Advances in communication technology and the emergence of new channels, such as social networks, have transformed the way people interact with each other. This shift towards online interactions has been taken into account in the analysis of GBV.

The study shows that more than half of the women surveyed reported having experienced online violence in the context of their relationship and online sexual violence. These findings replicate those described in other research conducted in similar geographic and age contexts, which found similar rates of victimization for online violence in relationships for women [22]. International studies have found similar results, indicating a high prevalence of victimization for women in intimate relationships, with social networks being the medium through which this form of violence is experienced [25,26].

The results of the study show that online violence is directly related to face-to-face violence. Online violence goes beyond the virtual realm; it has concrete and visible consequences for those who suffer it, and it often continues in physical settings [27]. Online violence is frequently the starting point of other forms of violence, likely due to the lack of direct contact with the victim [21]. Moreover, digital violence can be understood and recognized as a form of psychological violence. Therefore, it is essential to address online violence against women as a reflection of physical violence, and as a mirror of the society and culture in which such violence occurs—namely, as a result of the socialization of individuals within a patriarchal society [28].

The study shows a high rate of victimization by sexual violence. Of the total, 66% identified having suffered these forms of violence. Results for the macro survey of violence against women in Spain published in 2019 presents that it was 8.9% of women in the national territory who claimed to have suffered this form of violence and 40.4% claimed to have suffered sexual harassment, having 49.3% of women surveyed who claimed to suffer sexual violence or sexual harassment [29]. The analysis of these forms of violence is complex and results with different prevalence can be found due to the way in which the forms of violence analyzed are identified. In this case, we found that if we look at sexual violence as a group of in-person violence and sexual harassment, the results are similar to those found in the research.

The present study does not show significant differences between age groups regarding online violence against women. However, a study conducted in 2021 by the Belgian Institute for the Equality of Women and Men revealed a decrease in online violence as women’s age increases: 23% of women under the age of 25 had been victims, compared to 15% in the 25 to 34 age group, and approximately 8% among those over 35 [30].

GBV victimization at an early age is a major risk factor for experiencing more severe forms of violence later in life. It is also a factor in the development of other serious health conditions such as substance use, mental health problems, and dual pathology [31,32]. Femicide, the most extreme form of GBV described, occurs primarily in the adult population. For the year 2023, the Ministry of Equality reported that a total of 58 women had suffered femicide. The age group most affected by this violence was women between the ages of 31 and 40, followed by those between the ages of 41 and 50 [33]. Addressing online and in-person violence in the early stages of life can help to reduce GBV experienced in adulthood and lower the incidence of the most extreme form of GBV, femicide.

GBV is a multicausal phenomenon influenced by a variety of interrelated variables that either increase or decrease the likelihood of being a victim of this form of violence. Socioeconomic status appears to have an impact on a range of health and social problems, including GBV [34]. Victimization rates for direct online violence, online sexual violence, and in-person sexual violence are higher among women of lower socioeconomic status. A study in a sample of women from 28 European Union member countries found that those participants who identified themselves as having a low or poor socioeconomic status were at a greater risk of experiencing GBV from their current or former partners and were more likely to be worried about experiencing this form of violence [35]. Focusing on socioeconomic status in the fight against GBV is a key area of interest for researchers. Recent research has shown how an increase in women’s socioeconomic status is associated with a reduction in the risk of exposure to GBV [36]. Similarly, gender-based violence is a multifactorial phenomenon influenced by multiple interrelated variables, not solely by socioeconomic status. Various studies have shown that women with fewer resources are more likely to experience direct violence, as well as sexual violence both in person and online. However, to fully understand this reality, it is essential to incorporate an intersectional approach that examines how factors such as poverty, gender, ethnicity, disability, sexual orientation, and migration status intersect, giving rise to specific and exacerbated forms of violence. This perspective does not view inequalities as isolated elements, but rather as simultaneous axes that shape differentiated experiences of exclusion, lack of protection, and vulnerability. Therefore, a deeper analysis of violence against women must take into account the various forms of oppression that operate together and structurally [37].

Moving on to the associated factors that could explain the alarming rates of GBV, it has been observed that higher levels of pornography consumption are associated with higher rates of victimization for violence against women. Pornography use is greater among young and adult men. Greater exposure to this content among men may be related to a higher incidence of perpetrating violence, both on social networks and in person. A study conducted in a young Spanish population nationwide found that men who used pornography on a regular basis were more likely to engage in online violence within the context of their relationship [38]. At the same time, exposure to pornography has been linked to increased perpetration of forms of in-person violence in relationships, such as emotional abuse or sexual violence [39]. Rather than being a liberating tool in sexual relationships, pornography and its consumption among young people seem to contribute to the perpetuation of the subordination of women to men and reinforce the notion of heteropatriarchal superiority of men over women [40,41].

The final variable analyzed in relation to GBV was the amount of time participants spent on social networks. It was found that women who spent more time on social networks experienced higher levels of GBV. As the amount of time spent on social networking sites increased, so did the score for GBV experienced. These results are consistent with those obtained in studies analyzing online violence within relationships, reinforcing the connection between the amount of time spent on social networks and the level of violence experienced [13].

Correlations have been found between the different forms of violence analyzed. GBV, like other forms of violence, occurs on a gradual scale. In intimate relationships, violence typically begins with less severe forms of aggression, such as verbal or emotional abuse, and is followed by more severe forms of violence, such as physical or sexual violence and even femicide [42]. Early intervention in low-level violence is one of the most effective ways to break the cycle of violence and prevent more severe forms of GBV within relationships.

### Limitations

Several limitations must be considered when interpreting the results and the extrapolability of the data. First, the sample was selected through non-probability and purposive sampling, which limits the generalizability of the findings. Additionally, as the study relied on self-administered questionnaires, there is a potential for response bias or underreporting, particularly given the sensitive nature of the experiences explored. This may have led to an underestimation of the prevalence or severity of the phenomena analyzed.

One major limitation is the age range of the participants, which included only adults and young adults. Most of whom were university-aged samples, which could affect the extrapolation of results to the general Spanish population. Furthermore, this excludes the adolescent population, a group that often exhibits higher rates of social media use and, consequently, greater risk of exposure to online violence. Including adolescents in future research could yield valuable insights.

Furthermore, the complexity of the sexual violence reported suggests a likelihood of response bias or underreporting, which may have influenced the accuracy of the data. Although the analyses establish correlations between variables, they do not allow for causal inferences. To determine causality between variables, prospective longitudinal studies with regression analysis would be necessary to determine the timing of the variables’ appearance. In current research, it is unclear which variable precedes which other. However, identifying these associations is valuable for designing specific, effective interventions and for future studies that delve deeper into the causality of the variables analyzed here.

Finally, the study focused exclusively on in-person sexual violence, without addressing other forms of partner violence such as physical or psychological abuse. This methodological choice was made to ensure a concise and manageable questionnaire, and to prioritize a form of violence that is particularly severe, underreported, and closely associated with digital dynamics. However, we recognize that including a broader spectrum of violence could have enriched the analysis, and we recommend that future studies consider exploring these additional dimensions.

## 5. Conclusions

The study shows how a substantial part of the participants suffered the different forms of violence analyzed. This highlights the need to continue researching this problem due to its current scope. One of the ways to improve the fight against this event is to identify the variables that influence it. Either by identifying risk or protective factors that help us to make more effective interventions that achieve a decrease in the levels of violence against women. The study showed that the levels of violence experienced are higher among lower socioeconomic groups, those who spend more time on social media, and those who consume pornography. Although these results cannot establish causality, they do provide relevant information for developing specific programs and future research.

One of the most important findings was the correlation between the different forms of violence analyzed. Notably, there was a positive correlation between online sexual violence and in-person sexual violence. Violence against women is not confined to a single area or context. Experiencing violence in social networks or online is a major factor associated with in-person victimization. Addressing online violence could play a critical role in combating GBV in all other settings.

## 6. Patents

### Future Lines of Research

The research highlights the importance of addressing sexual violence against women not only from a physical perspective, but also from an online perspective, and the relationship we find between these two forms of violence. It also shows how levels of victimization also vary depending on socioeconomic status, time spent on social media, and pornography consumption. This opens the door to integrating these variables into future research and intervention programs, understanding these as possible associated risk factors.

## Figures and Tables

**Table 1 healthcare-13-02574-t001:** Sociodemographic characteristics.

Variables	n	(*%*)
**Age group**		
*Young adults*	277	23.5
*Adults*	900	76.5
**Socioeconomic level (euros per annum)**		
*Low (less than 30,000)*	779	66.2
*Medium (between 30,001 and 60,000)*	368	31.3
*High (more than 60,001)*	30	2.5
**Highest level of educational attainment**		
*No education*	1	0.1
*Primary education*	7	0.6
*Compulsory secondary education (ESO)*	22	1.9
*Baccalaureate or equivalent*	141	12.0
*Intermediate vocational training*	140	11.9
*University studies*	528	44.9
*Postgraduate studies*	338	28.7
**Social network usage time/day**		
*Less than 1 h per day*	88	7.5
*Between 1 and 3 h per day*	717	60.9
*Between 3 and 5 h per day*	306	26.0
*More than 5 h per day*	66	5.6
**Pornography consumption/viewing**		
*Yes*	231	19.6
*No*	945	80.3
**Frequency of pornography viewing**		
*Once or several times per year*	61	26.2
*Once or several times per month*	122	52.6
*Once or several times per week*	41	17.7
*Daily*	8	3.4

**Table 2 healthcare-13-02574-t002:** Violence experienced on social networks.

	Experience of Violence %	No Experience of Violence %
**Violence experienced online**		
*Overall*	68.2%	31.8%
*Direct aggression*	42.2%	57.8%
*Control*	65.0%	35.0%
**Sexual violence experienced online**	62.7%	37.3%
**Sexual violence experienced in person**	66.0%	34.0%

**Table 3 healthcare-13-02574-t003:** Differences in violence experienced as a function of age group.

	Young Adults	Adults		
	*Me*	*Me*	*U* de Mann–Whitney	*p* Value
**Violence experienced online**				
*Overall*	23.0	24.0	119,767.5	0.315
*Violence—control*	11.0	12.0	115,639.0	0.062
*Violence—direct*	11.0	11.0	119,984.5	0.293
**Sexual violence experienced online**	2.0	2.0	123,111.0	0.749
**Sexual violence experienced in person**	1.0	2.0	112,929.5	0.015 *

* Correlation significant at 0.05, *Me* = Median.

**Table 4 healthcare-13-02574-t004:** Violence experienced as a function of socioeconomic status.

Socioeconomic Status	Low	Medium	High		
	*Me*	*Me*	*Me*	Kruskal–Wallis	*p* Value
**Violence experienced online**					
*Overall*	24.0	23.0	26.0	5.3	0.072
*Violence—control*	12.0	11.0	14.5	5.3	0.069
*Violence—direct*	11.5	11.0	11.0	7.5	0.024 *
**Sexual violence experienced online**	2.0	1.0	1.0	11.1	0.004 **
**Sexual violence experienced in person**	2.0	1.0	1.5	13.7	0.001 **

* Correlation significant at 0.05, ** Correlation significant at 0.01, *Me* = Median.

**Table 5 healthcare-13-02574-t005:** Violence experienced as a function of time spent using social networks.

Time Spent Using Social Networks	Less Than 1 h/day	Between 1 and 3 h/day	Between 3 and 5 h/day	More Than 5 h/day		
	*Me*	*Me*	*Me*	*Me*	Kruskal–Wallis	*p*
**Violence experienced online**						
*Overall*	23.0	23.0	25.0	26.5	10.04	0.018 *
*Violence—control*	11.5	12.0	13.5	14.5	19.93	0.000 **
*Violence—direct*	11.0	11.0	11.0	11.5	6.66	0.084
**Sexual violence experienced online**	0	2.0	3.0	2.5	13.36	0.001 **
**Sexual violence experienced in person**	1.0	1.0	2.0	2.0	23.95	0.000 **

* Correlation significant at 0.05, ** Correlation significant at 0.

**Table 6 healthcare-13-02574-t006:** Violence experienced as a function of pornography consumption.

Consumption of Pornography	Yes	No		
	*Me*	*Me*	*U* de Mann–Whitney	*p* Value
**Violence experienced online**				
*Overall*	25.0	24.0	99,649.5	0.037 *
*Violence—control*	13.0	12.0	100,648.5	0.060
*Violence—direct*	11.0	11.0	99,048.5	0.015 *
**Sexual violence experienced online**	3.0	2.0	93,514.5	0.001 **
**Sexual violence experienced in person**	2.0	1.0	104,342.5	0.286

* Correlation significant at 0.05, ** Correlation significant at 0.01, *Me* = Median.

**Table 7 healthcare-13-02574-t007:** Correlation analysis between the different forms of violence.

	Overall Violence Experienced Online	Sexual Violence Experienced Online	Sexual Violence Experienced in Person
**Overall violence experienced online**	1.000	0.390 **	0.463 **
**Sexual violence experienced online**	0.390 **	1.000	0.518 **
**Sexual violence experienced in person**	0.463 **	0.518 **	1.000

Spearman’s Rho, ** Correlation significant at 0.01.

## Data Availability

The data presented in this study are available on request from the corresponding author. They comprise the raw, anonymized survey responses and are not publicly available to ensure participant privacy.

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
