# Peer review of "Violence Against Women on Social Networks: A Descriptive Analysis"

_healthcare, 2025, doi:10.3390/healthcare13202574_

Round 1
Reviewer 1 Report
Comments and Suggestions for Authors
The study seeks to determine the prevalence of gender-based violence experienced through social networks among adult women in Spain and to analyze its association with in-person sexual violence. This is a clearly defined and relevant research question.
The topic is highly relevant given the growing concern about online violence and its consequences for women’s health and safety. The focus on adult women, rather than exclusively adolescents or students, fills a gap in the literature and contributes original insights. The inclusion of socioeconomic variables, social media usage, and pornography consumption as predictors of victimization further strengthens the study’s novelty.
Compared with previous studies, this manuscript makes three notable contributions:
It provides robust data from a large sample (n=1,177) of adult women.
It highlights the strong correlation between online violence and in-person sexual violence, reinforcing the need to consider digital and physical abuse as interconnected phenomena.
It integrates sociodemographic and behavioral variables, offering a more comprehensive perspective on the factors that increase vulnerability.
The overall design is appropriate, but several points need improvement:
The use of purposeful and non-probability sampling limits generalizability. This should be more explicitly acknowledged.
The decision to restrict the sample to heterosexual women should be justified more clearly.
Ethical safeguards are mentioned, but further detail on measures taken to protect participants' psychological well-being during and after the survey would strengthen this section.
The results are well presented, though at times descriptive. A sharper analytical focus would improve clarity. The discussion successfully links findings to prior literature but occasionally repeats results rather than critically expanding on them. The limitations are well addressed but could be elaborated with greater emphasis on issues of representativeness and causality.
The conclusions are consistent with the evidence presented and address the main research question. However, they could be sharpened by more clearly articulating the study's unique contributions—particularly the link between pornography consumption, socioeconomic status, and vulnerability to gender-based violence. It would also be helpful to distinguish more explicitly between theoretical implications and practical recommendations for prevention and policy.
The reference list is appropriate and includes recent and relevant sources. Adding a few more international references on interventions to prevent violence against women would further strengthen the article.
This manuscript makes a valuable contribution to the field by documenting the prevalence of online gender-based violence and its connections with offline abuse. With revisions aimed at clarifying the methodology, sharpening the discussion, and enhancing the presentation of results, the paper has the potential to be an important addition to the literature on violence against women.
Author Response
First of all, I would like to thank you on behalf of all the researchers involved in this study. Your reviews help us improve the article.
The sections in the text that we hope will answer each of the questions have been indexed in yellow. In the updated Word document, you can check the comments section for the answers to each of the points mentioned for improvement.
We hope we have addressed the deficiencies in the research.
Kind regards.

Reviewer 2 Report
Comments and Suggestions for Authors
COMMENT 1: The word ‘pornography’ is not mentioned at all in the introduction to the article – appearing first under the methods section where we are told (p. 3 of 14) that participants were asked “how often they used online pornography”. While an association has been found between pornographic viewing and GBV, this association has been largely attributed in the literature to the potential for pornography (especially violent and coercive pornography) to be associated with coercive, abusive and violent behaviors by males in the context of intimate relationships. As such, could the authors please address the issue of pornography as a risk factor for GBV in the introduction, and provide a clear rationale for including pornographic usage in this female sample as a potential risk factor for GBV. In other words, it would be helpful for the authors to share with the readers why pornographic exposure by male partners was not assessed in the study, and what the rationale was for relying exclusively on pornographic exposure by female participants
COMMENT 2 (5 of 14, Table 1): In terms of 2024 OECD data approximately 42% of the Spanish population has some degree of tertiary education (74% for participants in this study) with approximately 20% of the population falling into the low socioeconomic category (66% in this study). As such there would appear to be some respects in which the study sample may not be representative of the general population – this needs to at least be mentioned by the authors.
COMMENT 3 (3 of 14): According to the authors, because: “the main objective of the study is to address gender-based violence experienced through social networks, only participants with a heterosexual orientation were selected” (emphasis added). I find it difficult to understand the logic of this decision, particularly if viewed in the context of the following views expressed by the Council of Europe:
“LGBT+ people (lesbian, gay, bisexual, transgender and other people who do not fit the heterosexual norm or traditional gender binary categories) also suffer from violence which is based on their factual or perceived sexual orientation, and/or gender identity. For that reason, violence against such people falls within the scope of gender-based violence”
Could the authors please address this issue more clearly in the paper.
COMMENT 4 (Discussion section). In the discussion section, the authors describe study findings, many of which are consistent with findings reported in the available literature (although some were not). It would enhance the paper if the authors could:
- Highlight novel findings more clearly (in other words, what does this paper contribute to current understandings); and
- Consider more directly the implications of these novel understandings for future research, practice, and theory.
Author Response

(The authors gave the same response as above.)

Round 2
Reviewer 2 Report
Comments and Suggestions for Authors
No comment